# The Effects of Photoperiodic Transcription Factor OsPRR37 on Grain Filling and Starch Synthesis During Rice Caryopsis Development

**DOI:** 10.3390/plants14233690

**Published:** 2025-12-04

**Authors:** Hanbing Zhang, Siqi Tang, Funan Wei, Wubei Zong, Junbin Hou, Xu Ran, Jingjing Zhao, Jingxin Guo, Zhonghua Wang

**Affiliations:** 1College of Agronomy, Northwest A&F University, Yangling 712100, China; 18829353496@163.com (H.Z.); tangsiqi2024@163.com (S.T.); woshiwfn@nwafu.edu.cn (F.W.); houjunbin123@163.com (J.H.); xran0188@gmail.com (X.R.); 18193959005@163.com (J.Z.); 2College of Bioscience and Biotechnology, Yangzhou University, Yangzhou 225009, China; wubei@yzu.edu.cn; 3College of Life Sciences, South China Agricultural University, Guangzhou 510642, China

**Keywords:** *Oryza sativa*, OsPRR37, grain filling, starch, grain quality

## Abstract

Grain filling governs grain weight formation in rice, while starch biosynthesis during this process critically determines both grain quality and yield. In this study, we characterized the heading date regulator OsPRR37 on regulating grain development, starch metabolism, and starch physicochemical properties. The *osprr37* mutants exhibited undesirable agronomic traits, including reduced plant height, decreased grain thickness, lower 1000-grain weight, and diminished yield. Moreover, mutant endosperm displayed irregular starch packing, aberrant granules morphology, and decreased granule diameter. Impaired grain filling was observed in *osprr37* mutants with reduced grain filling rates, which coincided with elevated soluble sugar content and reduced starch accumulation during grain development. Simultaneously, the expression of starch synthesis-related genes (SSRGs) was significantly altered. *osprr37* mutants had decreased total starch and amylose content, leading to reduced starch crystallinity, lower structural order degree, and impaired gelatinization properties. Collectively, our results demonstrated that OsPRR37 functions as a key regulator of grain filling and starch biosynthesis, thereby determining starch composition and physicochemical properties that ultimately affect rice quality and yield.

## 1. Introduction

Rice is a critically important food crop, feeding over 50% of the global population with approximately 155.5 million hectares cultivated [1]. Starch constitutes the primary component of rice grain, representing 80–90% of its dry weight. The starch located in the endosperm consists of amylose and amylopectin, the relative properties of these polymers critically influence both yield and quality [2]. Protein constitutes the second largest component, accounting for 7–10% dry weight in endosperm [3]. Minor constitutes include lipids, amino acids, vitamins, and other compounds [4]. Consequently, comprehensively identifying and harnessing starch biosynthesis regulators represents a critical strategy for developing elite rice varieties with enhanced quality [5].

Rice quality comprises four principal indices: processing quality, appearance quality, eating and cooking quality (ECQ), and nutritional quality [6]. Brown rice rate, head rice rate, and whole head rice rate are key indicators for assessing processing quality [7]. Appearance quality is primarily related to grain morphology and chalkiness. Key morphological parameters include length, width, thickness, and length-to-width ratio. Chalkiness refers to opaque areas in grain, resulting from insufficient grain filling and disorganized starch granule packing [8]. Eating and cooking quality, a paramount aspect of rice quality, is primarily determined by the composition and physicochemical properties of starch. Key ECQ evaluation indicators include amylose content, gel consistency, gelatinization temperature, peak viscosity, and enthalpy [9]. Generally, rice with low amylose content exhibits high gel consistency, low gelatinization temperature, and high peak viscosity, resulting in superior ECQ [6]. Proteins and amino acids serve as key indicators of nutritional quality. Grain storage proteins comprise the albumin, globulin, glutelin, and prolamin, with glutelin representing the predominant storage protein in rice [10].

Rice quality is dictated by starch characteristics, including content, composition, and physicochemical properties. These traits are governed not only by starch biosynthetic enzymes but also by transcription factors that modulate starch pathway gene expression. RSR1, an AP2/EREBF family transcription factor, represses starch biosynthesis genes, with its loss-of-function mutant showing reduced starch content and shifted amylose-to-amylopectin ratio [11]. The NF-Y family is involved in storage substance accumulation and endosperm development in rice. NF-YB1 forms a complex with NF-YC12 and bHLH144 that directly activates *Wx* expression to modulate amylose content [12]. Furthermore, NF-YB1 directly targets the *OsYUC11* promoter, enhancing auxin biosynthesis to regulate grain filling and storage compound accumulation [13]. The NAC transcription factors OsNAC20 and OsNAC26 directly activate starch and storage protein biosynthesis genes, while OsNAC25 forms a positive feedback loop with OsNAC20/OsNAC26 to stabilize starch biosynthesis pathways and maintain normal substance accumulation [14,15]. OsbZIP60 maintains ER homeostasis during endosperm development. Its loss-of-function mutant disrupts proteostasis, triggering OsbZIP50-mediated unfold protein change (UPR) signaling that up-regulates chalkiness-related genes, thereby inducing grain chalkiness [16]. The bHLH transcription factor OsFIF3 represses *OsSUT1* and *FLO2* expression, elevating grain chalkiness and reducing grain weights [17]. Overexpression of *OsTFIIB5* decreased plant height, accelerated flowering time, and increased seed storage protein accumulation while suppressing starch biosynthesis and altering starch composition [18].

The PRR (pseudo-response regulator) family is one of the core components of plant circadian clock, which is widely involved in various physiological process such as plant development, stress response, and grain productivity [19]. Natural variation of *TaPRR1* in wheat correlates with yield traits, particularly 1000-grain weight. Specific haplotypes simultaneously increase grain weight, reduce plant height, and accelerate heading [20]. OsPRR37 functions as a photoperiodic flowering repressor under long-day condition, delaying heading and increasing plant height. Concurrently, OsPRR37 modulates panicle differentiation rate to increase spikelet number per panicle, ultimately enhancing grain yield [21,22]. While the PRR family are regulating flowering and yield, their direct roles determining grain quality remain unknown. Therefore, this study aimed to elucidate the role of rice photoperiodic transcription factor OsPRR37 in regulating rice yield and quality through a comprehensive comparison of wild-type and *osprr37* mutants, focusing on yield-related traits and starch characteristics. We demonstrated that OsPRR37 functions as a pleiotropic regulator, influencing heading date, grain size, and yield potential, thereby concurrently determining multiple quality traits including processing, appearance, eating and cooking, and nutrition quality, which provide a critical insight for the genetic improvement of rice grain quality and yield.

## 2. Results

### 2.1. Effect of OsPRR37 on Agronomic Traits

Under natural long-day conditions in Yangling, both the wild-type and *osprr37* mutants failed to head. However, under artificial short-day conditions, the *osprr37* mutants headed 5 days earlier than wild-type plants. Then, we investigated mature plant phenotype to assess the effects of *OsPRR37* on rice growth and endosperm development (Figure 1). Compared to wild-type plant, plant height in the *osprr37* mutants decreased by 11.7%, while flag leaf length increased by 27.8%. However, flag leaf width and panicle neck length showed no significant differences. Yield components were compared between wild-type and the *osprr37* to assess the effect of *OsPRR37* mutation on agronomic traits. Versus wild-type, panicle length, spikelet number per panicle, number of secondary branches, and grain density increased significantly in the *osprr37* by 9.9%, 39.2%, 86.3%, and 31.2%, respectively. Panicle number per plant decreased by 31.2%, effective panicles per plant by 33.3%, and seed setting rate by 27.3%. Effective spikelet number per panicle and number of primary branches were slightly higher in the *osprr37* than wild-type (by 1.85% and 6.1%, respectively). Consequently, grain yield per plant in the *osprr37* mutants decreased significantly by 16.1% compared to wild-type.

### 2.2. Effect of OsPRR37 on Seed Morphology

To investigate the grain differences between wild-type and *osprr37* mutants, we quantitatively analyzed the grain morphology of brown rice, including grain length, grain width, grain thickness, and grain weight (Figure 2). The results showed that, compared to wild-type, the grains of *osprr37* mutants are longer (+5.46%) and wider (+2.68%), but these differences are not significant (*p* > 0.05). However, the grain thickness decreased significantly by 9.82% in *osprr37* mutants (*p* < 0.01). Transverse sections of *osprr37* mutant grains revealed chalky endosperm characterized by distinct opacity. The central endosperm region exhibited pronounced opacity, while the peripheral zone remained translucent, indicating severe chalkiness. This abnormal chalkiness was associated with incomplete grain filling, correlating with a 13.54% reduction in 1000-grain weight (*p* < 0.01).

### 2.3. The Rice osprr37 Mutants Display Defects in Grain Filling

To determine the effect of *OsPRR37* on grain development, wild-type and *osprr37* mutant plants were sampled at various grain developing stages. Grain dry weight and filling rates were analyzed to access developmental differences (Figure 3). The results showed that endosperm opacity in *osprr37* mutants progressively intensified during grain filling (Figure 3A). Figure 3B presents endosperm dry weight dynamic in wild-type and *osprr37* mutant grains during grain filling stage. Dry weight accumulation occurred rapidly during early grain filling stage (5–15 DAF, days after flowering), followed by progressively slower accumulation until plateauing at physiological maturity (~30 DAF). Throughout endosperm development, *osprr37* mutants accumulated 13.54% less dry weight than wild-type grains.

Analysis of grain filling rates revealed that the *osprr37* mutants exhibited consistently lower rates than wild-type during early grain filling stage (2–18.4 DAF) (Figure 3C). From 18.4 DAF onward, grain filling rates in the *osprr37* mutants converged with those of wild-type plant. Although both genotypes reached peak filling rates (Gmax) at approximately 6.5 DAF, the mutants maintained significantly reduced average filling rate ($\bar{G}$) throughout the grain filling period (17% lower than wild-type) (Table 1). These results indicated that *osprr37* mutants reduces early grain filling rates, contributing to the compromised carbon metabolism and decrease in final grain weight observed in mutants.

### 2.4. Effect of OsPRR37 on Carbohydrate Accumulation

We systematically profiled carbohydrate metabolism in developing rice endosperm by quantifying fructose, sucrose, total soluble sugar, and starch accumulation in wild-type and *osprr37* mutants at critical grain filling stages (5, 10, 15, 20, 25, 30 DAF) (Figure 4). Fructose content accumulation peaked during early grain filling (5–10 DAF) before declining rapidly. At 5 DAF, *osprr37* mutants exhibited 21.8% higher fructose content than wild-type (*p* < 0.01), with no significant differences observed during subsequent developmental stages (10–30 DAF) (Figure 4D). Compared with wild-type, the *osprr37* mutants accumulated significantly higher total soluble sugar levels during early-to-mid grain filling stages (5–20 DAF), with differences reaching 18–27% (*p* < 0.05). However, no significant differences were observed at late grain filling stages (25–30 DAF) (Figure 4C). Sucrose dynamics paralleled total soluble sugar patterns, progressively declining throughout grain development. The *osprr37* mutants consistently maintained elevated sucrose levels relative to wild-type until 30 DAF, as quantified in Figure 4B. Starch accumulation patterns revealed significant divergence between genotypes (Figure 4A). Wild-type plants reached the accumulation plateau at 20 DAF, with starch accounting for 76.66% of grain dry weight. In contrast, *osprr37* mutants exhibit delayed starch accumulation, reaching the plateau at 25 DAF and containing 71.13% starch. These results indicated that impaired sucrose-to-starch conversion in *osprr37* elevates soluble sugars content and decreases starch accumulation, directly reducing grain weight.

### 2.5. The Structure of Starch Compound Granules Is Abnormal in osprr37 Endosperm Cells

Scanning electron microscopy (SEM) of mature grain cross-sections revealed differences in endosperm architectures (Figure 5A–C). Wild-type endosperm displayed tightly packed starch granules with sharp edges and regular polyhedral structures. In contrast, *osprr37* mutant seeds exhibited loosely packed, irregularly spherical granules with extensive intergranular spaces through the central and peripheral regions. These structural irregularities promoted multiple light scattering events, resulting in the opaque phenotype of mutant grains.

### 2.6. Effect of OsPRR37 on Starch Particle Size

Starch granule fractions were categorized into three size classes: small (≤2 μm), medium (2–5 μm), and large (≥5 μm). Starch granule volume distributions are shown in Figure 6. Both wild-type and *osprr37* mutants exhibited unimodal volume distributions, with peak at 5.04 μm. Granule morphology analysis revealed significant alteration in *osprr37* (Table 1). Compared to wild-type, *osprr37* mutants exhibited 2.87% reduction in mean granule size and 2% decrease in volume-weighted mean diameter (both *p* < 0.05), while the 2.43% reduction in surface area-weighted mean diameter was non-significant (*p* > 0.05). Granule size distribution analysis revealed a 4.40% increase in the proportion of small starch granules, the proportion of medium starch granules increased significantly by 7.89% (*p* < 0.01), while large starch granules decreased significantly by 5.13% (*p* < 0.05) (Table 2). Overall, the *OsPRR37* mutation shifted starch distribution from large to small and medium sizes, generating significantly smaller granules. These findings demonstrated that *OsPRR37* mutation alters starch granule morphology.

### 2.7. Effect of OsPRR37 on Rice Grain Quality

The changes in rice processing and appearance quality were analyzed. Regarding processing quality, *osprr37* exhibited a significant 26.49% (*p* < 0.01) decrease in whole head rice rate compared to wild-type (Figure 7A). However, its brown rice rate increased 2.04% (*p* > 0.05); this difference was not statistically significant (Figure 7B). For appearance quality, the *OsPRR37* mutation increased chalky grain rate by 1.49-fold (*p* < 0.01) and chalkiness degree by 2.86-fold (*p* < 0.01) compared to wild-type (Figure 7C,D). In *osprr37* mutants, gel consistency decreased by 19.64% (*p* < 0.01), total starch content by 9.40% (*p* < 0.05), and amylose content by 9.26% (*p* < 0.05) compared to wild-type (Figure 7E–G). Rice endosperm contains four nutritional seed storage proteins (SSPs): albumin, glutelin, globulin, and prolamin. Quantitative analysis revealed significant SSP alterations in *osprr37* versus wild-type (Figure 7H–K): albumin increased 18.32% (*p* < 0.05), globulin decreased 12.69% (*p* > 0.05), glutelin increased 8.04% (*p* > 0.05), and prolamin increased 15.35% (*p* < 0.05). The significant albumin and prolamin increases versus non-significant globulin and glutelin changes suggested that *OsPRR37* loss-of-function significantly alters seed storage protein composition, with these compositional shifts potentially influencing rice quality traits.

Significant differences in solubility and swelling power were observed between wild-type and *osprr37* starches. *osprr37* starch exhibited significantly higher solubility and swelling power than wild-type (*p* < 0.05). At 95 °C, solubility increased from 10.44% to 12.89%, while swelling power rose from 21.03 g/g to 23.45 g/g (Figure 7L,M).

Starch gelatinization properties were analyzed using differential scanning calorimetry (DSC). DSC thermograms revealed similar gelatinization profiles between wild-type and *osprr37* mutant starch, though the mutant exhibited a slight reduction in gelatinization entropy (Figure 7N). Thermal characteristics are present in Table 3. Compared to wild-type, *osprr37* exhibited reductions in onset (*T*_0_), peak (*T_p_*), and terminal (*T_c_*) temperature (1.39%, 2.37%, and 2.43%, respectively) with a 12.89% decrease in gelatinization enthalpy (∆*H*). ∆*H* reflected energy consumption during starch granule dissolution, where higher values indicate great energy requirements. The 12.89% reduction in ∆*H* suggests lower crystalline perfection in mutant starch granules, aligning with observed morphological defects (Figure 5). These findings indicated that *osprr37* reduces rice starch gelatinization properties and enhances dissolution susceptibility, potentially due to alteration in granules size, amylose content, and amylopectin structure.

Given the importance of pasting properties in determining rice eating and cooking quality, we analyzed viscosity characteristics using Rapid Visco Analyzer (RVA). While wild-type and *osprr37* starch exhibited qualitatively similar temperature-dependent viscosity profiles (Figure 7O), significant qualitatively differences emerged in key parameters (Table 4). Wild-type starch exhibited characteristic viscosity development; viscosity increased with temperature until peak viscosity was attained. During subsequent cooling, viscosity briefly decreased before rapidly increasing to final viscosity. In contrast, *osprr37* mutant starch maintained a depressed viscosity profile throughout gelatinization, with peak viscosity occurring significantly earlier. RVA analysis revealed that *osprr37* exhibited significant reductions versus wild-type in peak viscosity, trough viscosity, breakdown, final viscosity, setback, and peak time. These parameters significantly decreased by 28.06%, 25.98%, 31.15%, 21.19%, 15.57%, and 2.26%, respectively. In contrast, pasting temperature increased by 21.3% (all *p* < 0.05). This altered viscosity profile correlates with observed structural defects in compound granules, including reduced granule packing density and increased surface fissures. These results demonstrate that *OsPRR37* mutation impairs starch pasting properties and affects rice eating and cooking quality.

### 2.8. Starch Physicochemical Properties in osprr37 Grain

Starch crystallinity was analyzed using X-ray diffraction (XRD). Wild-type and *osprr37* starch exhibit similar XRD patterns, showing strong diffraction single peaks at 15°, 17°, 18°, and 23°; this pattern is characteristic A-type starch crystallinity (Figure 8A). The *OsPRR37* loss-of-function mutation did not alter characteristic A-type XRD pattern but caused reduced XRD peak intensity and a 28.25% decrease in relative crystallinity versus wild-type (*p* < 0.05). This result demonstrates that *OsPRR37* loss-of-function causes crystalline structure defects that impaired starch functional properties, explaining the mutant’s lower gelatinization enthalpy (12.89% decrease) and peak viscosity (28.06% decrease).

FTIR spectra revealed similar starch absorption profiles in wild-type and *osprr37*, demonstrating that the *OsPRR37* loss-of-function mutation does not alter starch’s basic chemical structure (Figure 8B). However, *osprr37* exhibited significant absorbance attenuation across the 800–1200 cm^−1^ region, with significant spectral deviations at 995, 1022, and 1045 cm^−1^. Compared to wild-type, the 1045/1022 cm^−1^ and 1022/998 cm^−1^ ratio significantly decreased by 8.36% and 9.64% in *osprr37*, respectively (*p* < 0.05). Collectively, these results demonstrate that *OsPRR37* loss-of-function disrupts double-helical ordering, reduces short-range molecular organization, and induces structural reorganization of amorphous domains and lamellar architecture.

### 2.9. The Expression Pattern of Starch Synthesis-Related Genes and Enzyme Activities

To elucidate the function of *OsPRR37* in starch biosynthesis within rice endosperm, we analyzed the expression profiles of starch synthesis-related genes (SSRGs) in wild-type and *osprr37* endosperms using qRT-PCR at 5, 10, 15, 20, 25, and 30 DAF (Figure 9). Expression of *OsAGPS1* and *OsAGPL1*, which synthesize the starch precursor ADP-glucose, was consistently lower in *osprr37* than in wild-type. *OsPUL* transcripts were only detectable at low levels in the mutant at 5 DAF and 25 DAF. Moreover, several other starch synthesis genes (*OsAGPS2b*, *OsAGPL2*, *OsSSI*, *OsSSIIa*, *OsSSIIIa*, *OsGBSSI*, *OsSBEIIb*, *OsISA1*, and *OsISA2*) showed elevated expression in *osprr37* during mid-to-late developmental stages (15–30 DAF). Sucrose transporter genes were also affected, *OsSUT1* expression was reduced throughout development, while *OsSUT4* was transiently elevated at 20 DAF but otherwise suppressed.

To further verify these differences at the enzyme activity level, we quantified the activities of AGPase, GBSS, SSSase, and SBE in developing grains from 10 to 25 DAF. As presented in Figure 10, although the fluctuation in enzyme activities was minimal during grain filling, *osprr37* mutants consistently exhibited significantly reduced activities of four enzymes compared to wild-type. Specifically, throughout the grain filling period, AGPase activity in *osprr37* mutants decreased by 17.2–21.4%, GBSS activity decreased by 15–26.6%, SSSase activity decreased by 19.8–21.3%, and SBE activity decreased by 18.2–25%. Collectively, these alterations in both the activities of starch synthesis enzymes and the expression levels of their coding genes disrupt starch metabolism in *osprr37* endosperm.

## 3. Discussion

### 3.1. OsPRR37 Controls Rice Grain Filling and Starch Biosynthesis During Caryopsis Development

Previous study indicated that grain size is a primary determinant of grain weight, which ultimately determines grain yield [23,24,25,26,27,28]. However, grain filling rate also plays an important role in this process [29]. Grain filling is the progress whereby photosynthetic products from leaves are transported and converted into storage compounds within grains. Its dynamics are thus critical in determining rice quality, including processing, appearance, eating and cooking, and nutritional quality [30,31,32,33]. An impaired grain filling rate results in slower dry matter accumulation and reduced starch content, consequently leading to a decreased head rice rate and increased chalkiness degree [29]. Grain filling rate (average and maximum) showed a significant correlation with grain weight. Based on grain filling parameters, the grain filling period is divided into early, middle, and late stages. In the middle stage, the average grain filling rate peaks, representing the most critical phase for grain dry matter accumulation. While the correlation between grain filling duration and grain weight is generally weak, grain filling rate proves more influential than duration [34,35].

In this study, no significant differences in grain filling duration were observed between the wild-type and *osprr37* mutants throughout grain developing stages. However, for the grain filling rate during the middle stage, the maximum and average grain filling rates were significantly reduced in *osprr37*. This impairment led to slower dry matter accumulation and ultimately reduced grain weight and yield per plant (Figure 1). During grain filling stages, *osprr37* grains accumulated significantly higher levels of sucrose, fructose, and total soluble sugar compared to wild-type, while starch content was reduced throughout grain development (Figure 4). Collectively, these results suggested that *OsPRR37* mutation impairs starch synthesis and disrupts compound granule formation in endosperm during grain filling stages.

It is well known that starch constitutes the primary component of grain weight. Consequently, grain filling centers on starch synthesis and accumulation [36]. Starch biosynthesis requires multiple enzymes. Key enzymes include ADP-glucose pyrophosphorylase (AGPase), soluble starch synthase (SSSase), starch branching enzyme (SBE), and starch debranching enzyme (DBE) [37]. Enzyme activity and gene expression in the starch biosynthesis pathway significantly regulate grain filling. Their reduction impairs grain filling rates and causes incomplete grain development [29]. Sucrose, synthesized photosynthetically in leaves, serves as the primary precursor for starch biosynthesis. This sucrose is unloaded into endosperm cells via either the symplastic or apoplastic pathway. Sucrose unloading directly influences grain filling rates and starch biosynthesis [29,38]. AGPase catalyzes the rate-limiting step in the starch biosynthesis pathway, producing the essential precursor ADP-glucose (ADPG). Consequently, mutation in AGPase-encoding genes typically disrupts grain filling, causing grain shriveling along with reduced grain weight and starch content [39]. Furthermore, AGPase activity is strongly and positively correlated with grain filling rate, starch accumulation, and final starch content [40]. Granule-bound starch synthase I (GBSSI), encoded by Waxy gene, catalyzes amylose synthesis in the endosperm using ADP-glucose as substrate. The elongation, branching, and structural refinement of amylopectin chains are catalyzed by the coordinated action of soluble starch synthase (SSSase), starch branching enzyme (SBE), and starch debranching enzyme (DBE) [2]. SSSase catalyzes the elongation of α-1,4-glycosidic chains, and its activity level reflects the capacity for starch synthesis from ADPG. SBE introduces branches points by forming α-1,6-linkages and is essential for amylopectin synthesis [41].

Throughout the grain filling period, the *osprr37* mutants exhibited a slower grain filling rate and reduced starch accumulation, which corresponded to lower AGPase activity. This reduction in AGPase directly limited the supply of ADPG, the essential substrate for starch synthesis. Additionally, GBSS, SSSase, and SBE, all of which function as key downstream starch synthases, were consistently lower in *osprr37* mutants. This coordinated decline in enzyme activity indicated a reduced efficiency in utilizing ADPG for starch chain elongation and branching. Consequently, the overall capacity for starch synthesis was impaired, leading to the alterations in starch composition.

We found that the gene encoding AGPase, *OsAGPS1* and *OsAGPL1*, consistently exhibited low expression levels in *osprr37* mutants. In contrast, the transcript levels of *OsAGPS2b*, *OsAGPL2*, *OsSSI*, *OsSSIIa*, *OsSSIIIa*, *OsGBSSI*, *OsSBEIIb*, *OsISA1*, and *OsISA2* were higher than in wild-type during mid-to-late grain filling stages. This indicated that the loss of *OsPRR37* function may trigger a compensatory feedback mechanism that upregulated the expression of these genes. In barley, functional disruption of *HvSSIIa* led to the compensatory upregulation of other starch synthesis genes, such as *HvGBSSI*, *HvSSI*, and *HvSBEIIa*, to maintain starch synthesis [42]. During the grain filling stage, the endosperm of the *osprr37* mutants accumulated high levels of soluble sugars. The sugars function not only as metabolic substrates but also as signaling molecules that regulate the expression of starch biosynthesis genes [43,44]. Consequently, under these high-sugar conditions, the expression of several genes in the starch synthesis pathway was upregulated. The regulatory pattern is consistent with other cereals. In wheat *gbssi* mutants, sucrose and glucose contents were elevated, and the transcription abundance of *SSs*, *SBEs*, and *DBEs* was increased. Similarly, *sbeiia* mutants exhibited increased fructose and sucrose levels, along with upregulated expression of *GBSSI* and *ISAII*; *ISAII* participates in amylose synthesis by providing GBSSI with α-glucan chains for elongation [45]. An analogous regulation in transcription levels was also observed in rice *gbssi* mutants [46]. Despite the compensatory regulation of some starch synthesis genes in the mid-to-late grain filling stages, the dysregulated gene expression in *osprr37* mutants not only failed to compensate for the starch deficiency but also delayed starch synthesis initiation and altered starch composition and structure, ultimately reducing grain weight and yield per plant.

### 3.2. Effects of Assimilate Accumulation on Rice Quality Traits

Starch, protein, and soluble sugars constitute the primary components of endosperm, and their contents and interactions significantly influence rice quality [47,48]. Critically, grain chalkiness stems not only from deficient starch biosynthesis but also from disrupted protein–starch equilibrium during endosperm development [49]. When endosperm development is compromised, elevated protein synthesis causes protein deposition within inter-granules spaces, physically preventing dense packing of starch granules. This spatial competition creates structure voids in the endosperm, resulting in increasing light scattering, thereby causing chalkiness phenotype [50]. In addition, soluble sugars (e.g., sucrose) serve as substrates for starch biosynthesis in grains, and their conversion efficiency directly affects grain filling and starch accumulation [38]. Excess sucrose and fructose during late grain filling promote storage protein accumulation while disrupting starch granule arrangement; this disordered crystalline packing elevate chalkiness, reducing both processing and appearance quality [51,52].

In this study, compared to wild-type, *osprr37* mutants exhibited reduced starch synthesis capacity and impaired conversion of sucrose to starch, resulting in elevated sucrose content prior to 30 DAF and higher total soluble sugars from 5–25 DAF, while starch content remained consistently lower throughout the entire grain filling period. The metabolic imbalance in *osprr37* suppressed sucrose conversion and starch accumulation, which induced disorder starch granule packing and compromised grain appearance quality. Meanwhile, *osprr37* exhibited higher storage protein contents (especially glutelin, prolamin, and albumin) and more abnormal starch granules than wild-type. Excessive protein accumulation in the gaps between starch granules in *osprr37* may limit starch granule polymerization and reduce grain compactness, contributing to increased chalkiness and decreased head rice rate.

Starch is water-insoluble, but upon heating, granules swell in the amorphous region and form a viscous paste. This phase transition is gelatinization, with its characteristic onset temperature termed gelatinization temperature [53]. Rice starch gelatinization depends not only on starch composition but also on storage proteins. A protein network structure formed by storage proteins (notably glutelin and prolamin) encapsulates starch granules, physically limiting water access during heating. This impedes granule swelling and amylose leaching, thereby reducing peak viscosity during gelatinization [52,54,55]. Moreover, hydrophobic storage proteins exhibit strong competitive hydration capacity. Elevated protein content reduces free water availability for starch gelatinization and restricts water mobility, thereby increasing the activation energy required for dissolution of internal starch crystallites and significantly impeding gelatinization progress [56,57]. In this study, *osprr37* mutants enhanced the accumulation of glutelin, prolamin, and albumin, which formed a protein barrier that impeded starch gelatinization. This led to an increased pasting temperature and a decreased peak viscosity. Collectively, *OsPRR37* mutation altered grain composition—reducing starch while elevating soluble sugars and storage proteins—while collectively impairing milling yield, increasing chalkiness, and degrading appearance and eating and cooking quality.

In this study, the increased protein content in *osprr37* mutants suggests the improvement in nutritional quality. However, this benefit is offset by reductions in amylose and total starch content, which adversely affect other critical quality traits: processing, appearance, and eating/cooking quality. The decrease in total starch content resulted in powdery and friable grains and increased chalkiness, indicating poor processing and appearance quality. Concurrently, the decrease in amylose content led to undesirable alterations in pasting and gelatinization properties, negatively influencing eating and cooking quality such as rice stickiness, elasticity, and overall palatability. Consequently, despite the nutritional enhancement, the overall grain quality is substantially compromised.

Furthermore, we observed similar phenotypes in other crops as well. The maize Opaque-2 (O2) protein is a key regulator of endosperm development and carbon–nitrogen metabolism, whose loss of function leads to significantly reduced starch content and yield [58,59]. In our study, *osprr37* mutants exhibit a parallel phenotype: disrupted starch metabolism, elevated soluble sugars, reduced starch content, increased protein content, and lower yield. These key traits are highly similar to the phenotypes of the maize *o2* mutants, suggesting that OsPRR37 may be similar to the O2 protein and play a regulatory role in grain yield and quality formation.

### 3.3. OsPRR37 Regulates Starch Molecular Architecture and Functional Properties

Starch gelatinization is an irreversible order–disorder transition initiated by moist heat, characterized by granule swelling, crystalline melting, and amylose leaching due to hydrogen dissociation [60]. Differential scanning calorimetry (DSC) characterized starch thermal properties via gelatinization enthalpy (∆*H*) and transition temperature (*T_o_*, *T_p_*, *T_c_*). Reduced amylose content typically correlates with decreased stability of the crystalline structure, consequently lowering the temperature required for gelatinization. In contrast, starch with high amylose content requires greater thermal energy to break hydrogen bonds in crystalline regions and exhibits reduced swelling capacity due to limited granule hydration, synergistically elevating gelatinization temperature [61]. Starch granule size further influences gelatinization properties: smaller granules gelatinize at low temperatures due to higher surface area–volume ratios, requiring less thermal energy for phase transition [62]. Gelatinization enthalpy (∆*H*) quantifies the endothermic disruption of double-helical crystallites. Since ∆*H* scales with crystallinity, highly crystalline starch requires greater thermal energy input to overcome its structure stability during transition [63,64]. Concurrent reductions in amylose content, gelatinization temperature, relative crystallinity, and gelatinization enthalpy, along with elevated swelling power and water solubility, indicate disrupted double-helical packing in *osprr37* starch granules. This structure instability facilitates phase transition at lower thermal energy inputs.

Alterations in starch molecular structure and composition directly determine pasting behavior [65]. *OsPRR37* mutation altered starch functional properties; key RVA parameters (peak viscosity, trough viscosity, breakdown, setback, peak time) decreased significantly, whereas gelatinization transition temperature increased, indicating modified hydrothermal stability. Peak viscosity (PV) measures the maximum paste resistance during heating, quantifying starch swelling power achieved through hydration-driven granule expansion prior to rupture [66]. Consistent evidence indicates that large starch granules exhibit greater water absorption capacity due to reduced surface area-to-volume ratios, consequently elevating peak viscosity through enhanced swelling power [67]. Final viscosity (FV) quantifies paste viscosity at 50 °C, serving as a key indicator of retrogradation tendency. High-amylose starch promotes robust inter-chain hydrogen bonding, accelerating amylose recrystallization during cooling. This strengthens gel networks and elevates FV [66]. The significantly lower PV and FV and in *osprr37* starch directly stem from its 18.7% amylose reduction and 32% small granule diameter (Figure 6 and Figure 7), collectively accelerating hydration while compromising structural stability during heating.

## 4. Materials and Methods

### 4.1. Plant Materials and Growth Condition

Rice materials of wild-type and *osprr37* mutants used in this study were provided by Professor Jingxin Guo of South China Agricultural University. Wild-type plants (with *Hd1*, *Ghd7*, *DTH8*, and *OsPRR37*) were selected from an F_5_ family from a cross between an *indica* landrace (accession no. I7) and the *indica* cultivar Qinghuazhan [68]. Their unpublished study generated *osprr37* mutants by creating a mutated *osprr37* allele at target site 5′-CTCATCACAACCAAACtCGCcgg-3′ via CRISPR/Cas9-mediated knockout of sequence 5′-CTCATCACAACCAAACCGCcgg-3′ in the first exon of *OsPRR37*. Rice plants were cultivated in the experimental greenhouse at Northwest A&F University, Yangling, Shaanxi Province, China (34°16′ N, 108°04′ E). From mid-May to October, plants were maintained under artificial short-day conditions (11–11.5 h day-lengths) using daily shading before dusk.

### 4.2. Determination of Agronomic Traits and Grain Filling Rate

To elucidate the process of rice yield formation, we systematically measured key agronomic traits at maturity, including plant height (PH), panicle length (PL), panicle number per plant (PNP), effective panicles per plant (EPP), spikelet number per panicle (SNPP), effective spikelet number per panicle (ESPP), seed setting rate (SSR), number of primary branches (NPB), number of secondary branches (NSB), flag leaf length (FLL), flag leaf width (FLW), panicle neck length (PNL), grain density (GD), and yield per plant (YPP). The grain morphological traits (grain length, width, thickness, and 1000-g weight) were measured after harvest using a seed test instrument (SC-G, Wanshen, Shenzhen, China). Seeds at different grain filling stages (5, 10, 15, 20, 25, and 30 days after flowering, DAF) were collected, oven-dried at 65 °C to constant weight, dehulled, and weighed. The Richards equation [W = A(1 + Be^−kt^)^−1/N^] was fitted to derive grain filling parameters [69].

### 4.3. Determination of Carbohydrate Content

To elucidate the physicochemical basis of yield and quality formation, we quantified sugar and starch content according to the method of Wang et al. [70]. Harvested grains were dehulled, dried to constant weight, and ground into powder. Next, 0.2 g rice flour was mixed with 10 mL deionized water and incubated at 100 °C for 30 min. After cooling to 25 °C, the mixture was centrifuged at 2000× *g* for 6 min. The supernatant was collected and adjusted to 40 mL as the extract for soluble sugar quantification. The dried residue was first hydrolyzed with 2 mL deionized water at 100 °C for 30 min, followed by the addition of 2 mL 9.2 M HClO_4_ for further incubation at 100 °C for 30 min. After cooling to 25 °C, 6 mL deionized water was added and vortex-mixed. Following centrifugation (2000× *g*, 6 min), the supernatant was retained. The precipitate underwent identical extraction with 4.6 M HClO_4_. Combined supernatants were volumetrically adjusted to 40 mL as the starch extract. Total soluble sugar content and starch content were quantified by the anthrone method. Sucrose content and fructose content were determined using the resorcinol assay [71].

### 4.4. Analysis of Starch Physicochemical Properties

To analyze key starch composition and function properties, we measured the amylose content, gel consistency, starch water solubility, and swelling power. Amylose content was quantified using iodine colorimetry [72]. Then, 20 mg rice flour was suspended in 100 μL anhydrous ethanol, homogenized with 1.8 mL 1 M NaOH, and digested at 60 °C for 1 h. After cooling to 25 °C, 100 μL precipitate was mixed with 9 mL deionized water and 200 μL 1 M acetic acid. Following the addition of 200 μL 0.02% I_2_-KI, samples were measured at 620 nm. Gel consistency was assessed according to the standard tube test method [73]. Next, 100 mg rice flour was mixed with 0.2 mL thymol blue indicator and 2.0 mL 0.2 M KOH. After vortex homogenization, samples were boiled for 8 min, cooled to room temperature, and chilled at 0 °C for 20 min, then gel length was measured immediately using a caliper. Starch solubility and swelling power were determined following Kong et al. [74]. Exactly 150 mg (W0) starch was suspended in 10 mL deionized water and vortex-mixed thoroughly. Samples were incubated at 55 °C, 65 °C, 75 °C, 85 °C, and 95 °C for 30 min with vortex mixing every 2 min. After cooling to 25 °C, centrifugation was performed at 3000× *g* for 20 min. The supernatant was dried to constant weight (W1). The wet precipitate (W2) was weighed, then dried to constant weight (W3). The swelling power (SP) and water solubility (WS) were calculated as follows: SP (g/g) = W2/W3, WS (%) = (W1/W0) × 100%.

### 4.5. Determination of Storage Protein Content

To evaluate the nutritional quality, grain storage proteins were extracted and quantified from rice flour according to the method described by Chen et al. [52]. Albumin, globulin, glutelin, and prolamin fractions were sequentially extracted from 1.0 g dehulled rice flour. Each extraction was performed at room temperature with continuous shaking of 2 h, followed by centrifugation at 10,000× *g* for 5 min to collect the supernatant. This extraction process was repeated twice, and the combined supernatant was diluted to a final volume at 25 mL. For protein quantification, 1 mL aliquots of each extract were mixed with 5 mL Coomasie Brilliant Blue G-250 (Bio-Rad Laboratories, Hercules, CA, USA), and absorbance was measured at 592 nm.

### 4.6. Analysis of Starch Granule Morphology and Size Distribution

To comprehensively characterize starch granule properties, we employed multiple approaches. Starch granule morphology was analyzed by field-emission scanning electron microscopy (FESEM, S-4800, Hitachi, Tokyo, Japan). Transverse sections of mature seeds were prepared using razor blades. The air-dried sections were mounted on aluminum stubs, gold-sputtered for conductivity, and imaged by FESEM. To examine the development of compound granules, transverse sections of developing endosperm harvested at 10 DAF were used to prepared semithin sections using a microtome (Leica, UC7, Nussolch, Germany). The 1 μm sections were stained with 1% I_2_-KI and imaged by light microscopy (Leica, MZ10F, Wetzlar, Germany). Starch particle size distribution was determined by laser diffraction particle size analyzer (Mastersizer 2000, Malvern Instruments, Malvern, UK). Then, 10 mg of starch was dispersed in 5 L of deionized water and homogenized by vortex mixing. Measurements were performed immediately to prevent sedimentation.

### 4.7. Starch Crystallinity Analysis

The crystalline structure and relative crystallinity of starch granules were analyzed using an X-ray diffractometer (XRD, D8 ADVANCE A25, Bruker AXS GmBH, Karlsruhe, Germany) according to the method of Guo et al. [75]. The operating parameters were set as follows: voltage 40 kV, current intensity 100 mA, scanning range 5–40° (2θ), step size 0.02°, and the scanning speed 10°·min^−1^. The relative crystallinity of starch was calculated using MDI Jade 6.0 software.

### 4.8. Fourier Transform Infrared Spectrum Analysis

To characterize the molecular order and helical arrangement of starch components at short-range scales through Fourier transform infrared spectrum (FTIR, Bruker Optics, Karlsruhe, Germany), starch samples were thoroughly blended with potassium bromide to make a slice and put into the sample chamber. FTIR spectra were acquired in the scanning wave number region between 4000 and 500 cm^−1^ at a resolution of 4 cm^−1^ over 42 scans. Absorbance values of starch at 1045 cm^−1^, 1022 cm^−1^, and 995 cm^−1^ were obtained from the deconvolution infrared spectra, and the variation of starch molecular order was analyzed by calculating the absorbance ratio at 1044/1022 cm^−1^ and 1022/998 cm^−1^.

### 4.9. Thermal Properties Analysis

To characterize the thermal behavior and gelatinization properties of starch, differential scanning calorimetry (DSC) was employed. Accurately weighed 5 mg of starch was placed in a crucible, mixed with 15 μL of deionized water, sealed, and equilibrated at 4 °C for 24 h. During analysis, an empty crucible was used as the reference sample, then the test samples were heated from 30 °C to 100 °C at a rate of 10 °C·min^−1^, with three independent replicates performed for each sample. The DSC curve was analyzed using the instrument’s own software (TA Universal Analysis Version 4.5, TA instruments, New Castle, DE, USA) to determine thermal characteristic parameters: onset temperature (*T_0_*), peak of gelatinization temperature (*T_p_*), conclusion temperature (*T_c_*), and gelatinization enthalpy (∆*H*).

### 4.10. Pasting Properties Analysis

The starch pasting properties were investigated using the Rapid Viscosity Analyzer (RVA4500, Perten, Stockholm, Sweden) according to the method of Guo et al. [75]. Rice flour (3 g) was weighed and thoroughly mixed with 25 mL of deionized water in an aluminum barrel. The procedure was as follows: equilibrium at 50 °C for 1 min, heating to 95 °C at a rate of 12 °C·min^−1^, holding at 95 °C for 2.5 min, cooling to 50 °C at the same rate, and maintaining at 50 °C for 2 min. The following parameters were derived from the pasting curve: peak viscosity (PV), trough viscosity (TV), final viscosity (FV), breakdown (BD), setback (SB), pasting temperature (PT), and pasting time (PeT).

### 4.11. RNA Extraction and Real-Time PCR Analysis

To analyze the expression profiles of starch synthesis-related genes, total RNA was extracted from rice seeds at various grain filling stages using the RNAprep Pure Plant Kit (DP441). RNA integrity was verified by agarose gel electrophoresis. cDNA synthesis was performed with ToloScript ALL-in-One RT EasyMix (#22107). Quantitative PCR used 2× Universal SYBR qPCR Master Mix (QP101_Ver.1) with *OsActin* (Os03g071810) as reference, following the manufacture’s protocol. Relative expression was calculated using the 2^−ΔΔCT^. Primer sequences are listed in Appendix A.

### 4.12. Enzyme Extract and Activity Assay

To investigate the enzymatic changes in starch synthesis, the activities of key starch biosynthetic enzymes (AGPase, GBSS, SSSase, SBE) were quantified in developing grains. Grains were collected at 10, 15, 20, and 25 DAF. Samples were immediately flash-frozen in liquid nitrogen and stored at −80 °C for subsequent analysis. For enzyme extraction, 1 g grains were homogenized in 9 mL pre-cooled extraction buffer as described by Wang et al. [76]. The homogenate was then centrifuged at 4 °C, 10,000× *g* for 30 min, and the supernatant was collected for enzyme activity assay. The AGPase, GBSS, SSSase, and SBE activities were measured using enzyme-linked immunosorbent assay kits (Shanghai FANKEL Industrial Co., Ltd., Shanghai, China, No. F7859-B, F50055-B, F7906-B, and F7907-B, respectively) following the manufacturer’s protocols.

### 4.13. Statistical Analysis

In this study, at least two replicates were performed in each experiment. The data were analyzed by SPSS (version 25.0, Chicago, IL, USA) and presented as mean ± standard (mean ± SD). The significant difference between samples was determined using Student’s *t*-test; * and ** indicate *p* < 0.05 and *p* < 0.01, respectively.

## 5. Conclusions

This study reveals how the photoperiodic transcription factor OsPRR37 regulates rice grain yield and quality. The *osprr37* mutants exhibited significant alteration in agronomic traits compared to wild-type. Plant height, panicles per plant, and seed setting rate were significantly reduced in the mutant. Grain thickness and 1000-grain weight decreased, concomitant with increased chalky grain rate and severe chalkiness phenotype. These defects were associated with impaired grain filling in *osprr37* mutants, evidenced by significantly reduced grain filling rates throughout the grain developing period. During the grain filling stage, starch biosynthesis was impaired in *osprr37* mutants, resulting in elevated soluble sugar contents, reduced starch accumulation, and dysregulated expression of SSRGs. Microstructural analysis revealed incomplete endosperm filling and aberrant starch granule morphology in *osprr37* grains. The *osprr37* mutants showed inferior processing and appearance quality, though it accumulated higher storage protein content. *OsPRR37* mutation reduced total starch and amylose content, consequently decreasing starch viscosity and gelatinization enthalpy. These changes correlated with reduced starch granule size, decreased crystallinity, and diminished molecular order in endosperm. Although we reveal the role of *OsPRR37* in regulating grain yield and quality, the precise molecular mechanisms underlying these effects remain largely elusive. Future research should focus on the following aspects: protein–protein interaction screening using yeast two-hybrid or Co-immunoprecipitation assay could identify functional proteins that bind to OsPRR37 and modulate its expression or activity; high-throughput technologies such as Chip-seq or DAP-seq could be employed to identify its downstream target genes. These approaches will elucidate the molecular mechanisms through which OsPRR37 regulates grain yield and quality.

## Figures and Tables

**Figure 1 plants-14-03690-f001:**
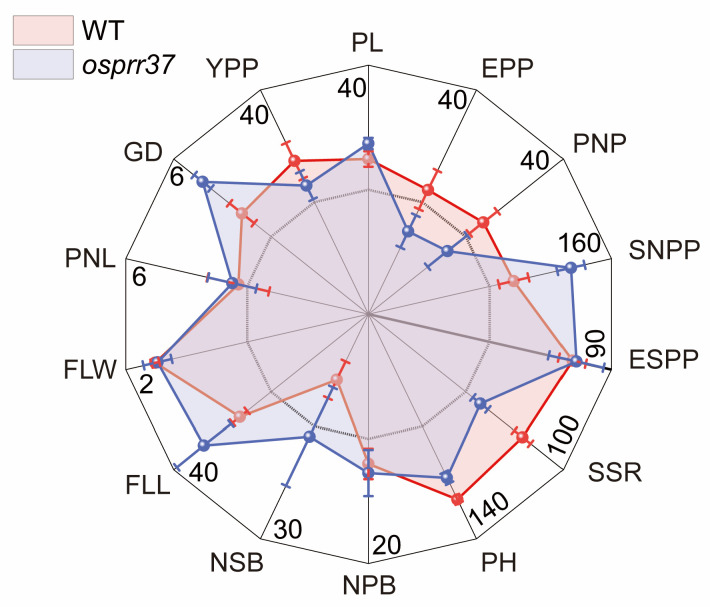
Agronomic characteristic comparison of wild-type (WT) and *osprr37* mutants. PL, panicle length; EPP, effective panicles per plant; PNP, panicle number per plant; SNPP, spikelet number per panicle; ESPP, effective spikelet number per panicle; SSR, seed setting rate; PH, plant height; NPB, number of primary branches; NSB, number of secondary branches; FLL, flag leaf length; FLW, flag leaf width; PNL, panicle neck length; GD, grain density; YPP, yield per plant.

**Figure 2 plants-14-03690-f002:**
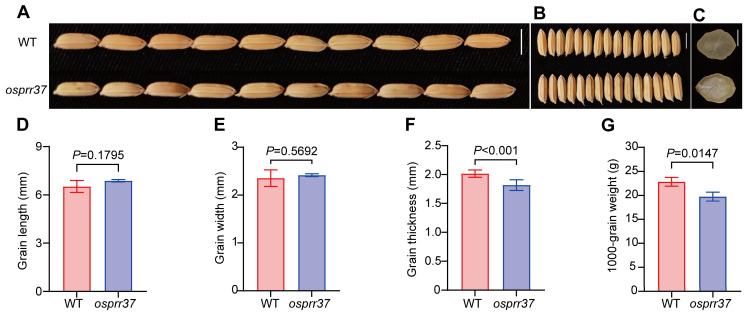
Phenotype comparison of wild-type and *osprr37* mutant mature grains. (**A**,**B**) Phenotype of grains for wild-type and *osprr37* mutants. (**C**) Cross-section of brown rice. (**D**–**G**) Examination of grain length, grain width, grain thickness, and 1000-grain weight. There were three independent biological replicates. Data indicate means ± SD from at least 3 biological replicates. Student’s *t*-test was used to generate the *p* values. Bar = 5 mm in (**A**). Bar = 4 mm in (**B**). Bar = 1 mm in (**C**).

**Figure 3 plants-14-03690-f003:**
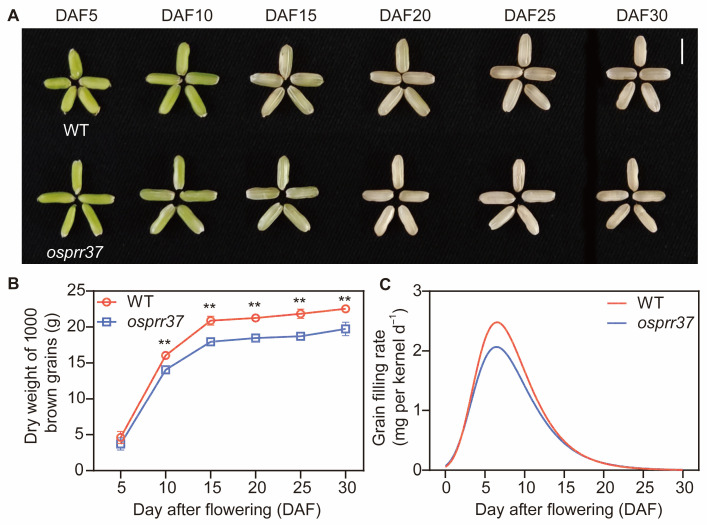
Dynamic changes of grains, endosperm dry weight, and grain filling rate in wild-type and *osprr37* mutants. (**A**) Grain phenotype, (**B**) endosperm dry weight, and (**C**) grain filling rate at different days after flowering; Data were analyzed using *t*-test and presented as means ± SD from 3 biological replicates. ** *p* < 0.01. Bar = 5 mm in (**A**).

**Figure 4 plants-14-03690-f004:**
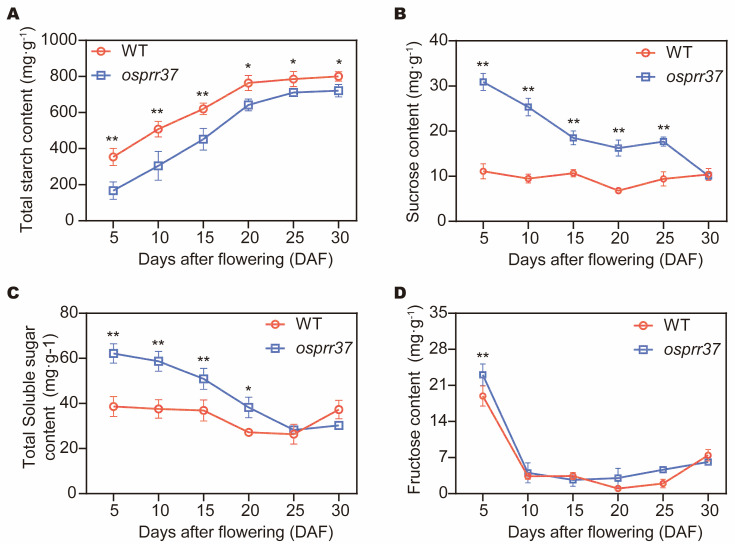
Dynamic changes of carbohydrate during grain filling process. The content of starch (**A**), sucrose (**B**), total soluble sugar (**C**), and fructose (**D**) in the grain of wild-type and *osprr37* mutants. Data were analyzed using *t*-test and presented as means ± SD from 3 biological replicates. * *p* < 0.05, ** *p* < 0.01.

**Figure 5 plants-14-03690-f005:**
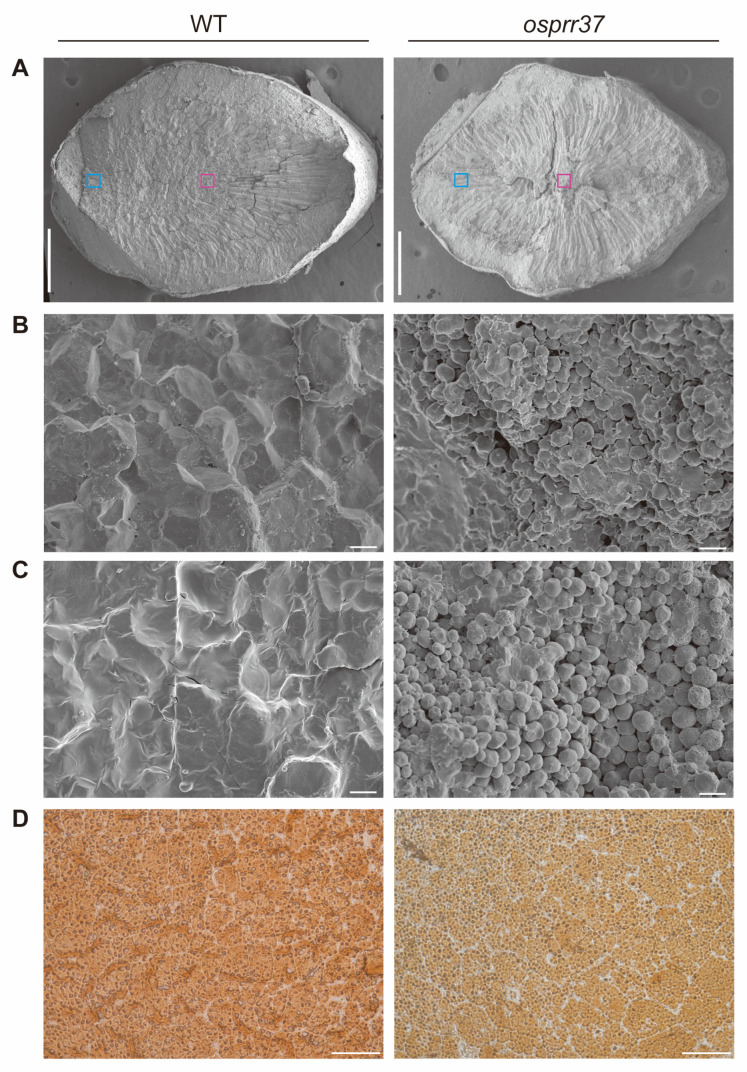
Starch granule morphology in endosperm cells. (**A**) SEM images of the transverse sections of mature endosperms from wild-type and *osprr37*. Bars = 5 mm. (**B**) SEM images of central region in mature endosperms from wild-type and *osprr37*. The areas shown in higher magnification are indicated by pink boxes. Bars = 50 μm. (**C**) SEM images of periphery region in mature endosperms from wild-type and *osprr37*. The areas shown in higher magnification are indicated by blue boxes. Bars = 50 μm. (**D**) Semithin sections of wild-type and *osprr37* developing endosperms at 10 DAF. Bars = 50 μm. Semithin sections of developing endosperms at 10 DAF were generated to examine the structure of starch compound granules in wild-type and *osprr37* (**D**). Wild-type endosperm cells contained abundant developed polygonal starch granules with uniform size and dense spatial packing. In contrast, *osprr37* exhibited incomplete granule filling with prominent endosperm voids and significantly fewer compound starch granules. Mutant endosperm also showed numerous structurally defective compound starch granules that were dispersed, fractured, loosely packed, and weakly strained, distributed among normal granules. These findings demonstrate that *OsPRR37* mutation inhibit standard compound starch granule formation.

**Figure 6 plants-14-03690-f006:**
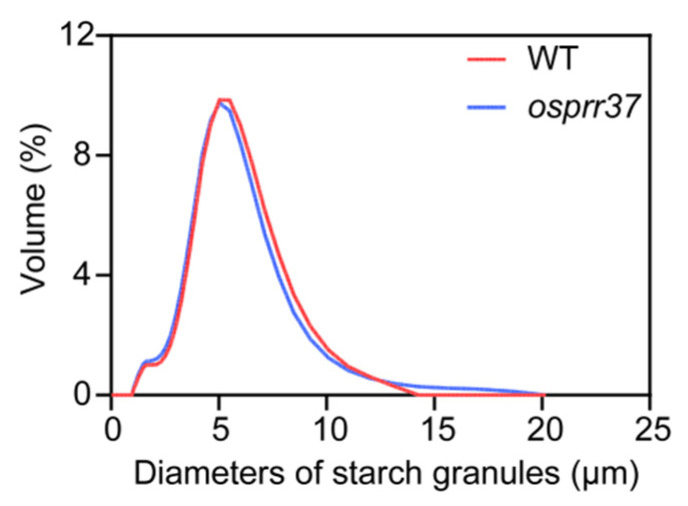
Distribution of particle size in endosperm starch granules.

**Figure 7 plants-14-03690-f007:**
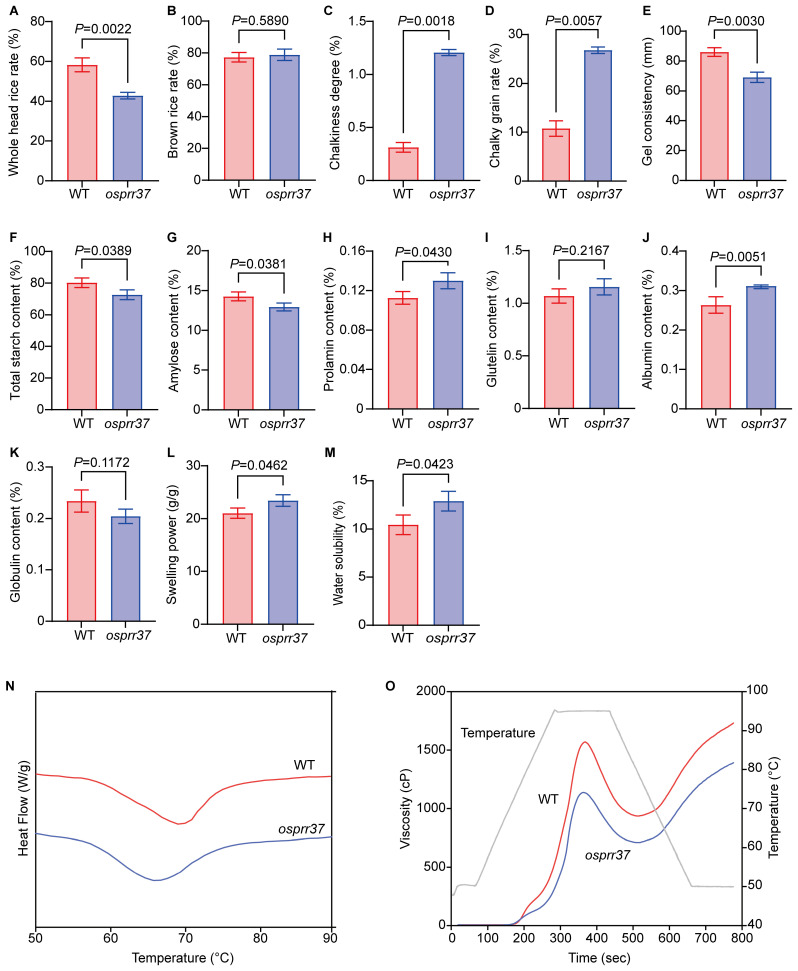
Grain quality traits in wild-type and *osprr37* mutants. (**A**,**B**) Grain processing quality; (**C**,**D**) grain appearance quality; (**E**–**G**) grain eating and cooking; (**H**–**K**) grain nutritional quality; (**L**) swelling power; (**M**) water solubility; (**N**) starch thermal property; (**O**) starch RVA pasting curves. Data indicate means ± SD from 3 biological replicates. Student’s *t*-test was used to generate the *p* values.

**Figure 8 plants-14-03690-f008:**
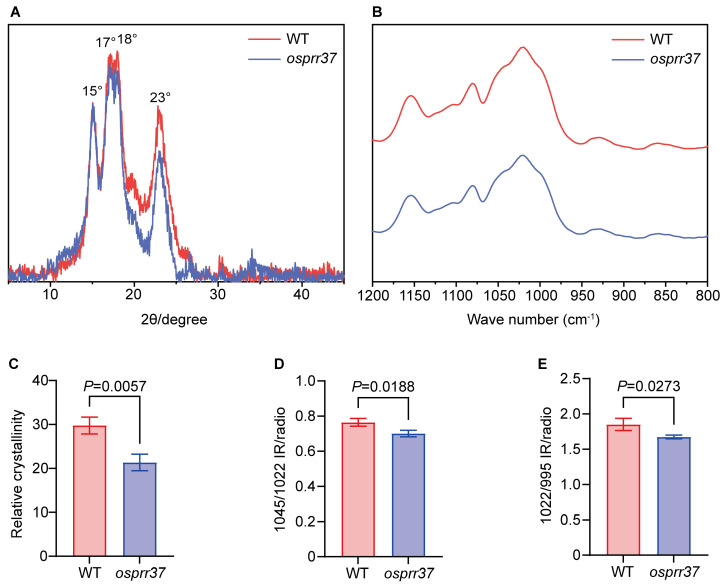
Starch crystal structure of wild-type and *osprr37* mutants. (**A**) XRD spectra analysis of starches; (**B**) Fourier transform infrared spectroscopy (FTIR) spectra analysis of starches; (**C**) relative crystallinity; (**D**) order degree of starch; (**E**) disorder degree of starch. Data indicate means ± SD from 3 biological replicates. Student’s *t*-test was used to generate the *p* values.

**Figure 9 plants-14-03690-f009:**
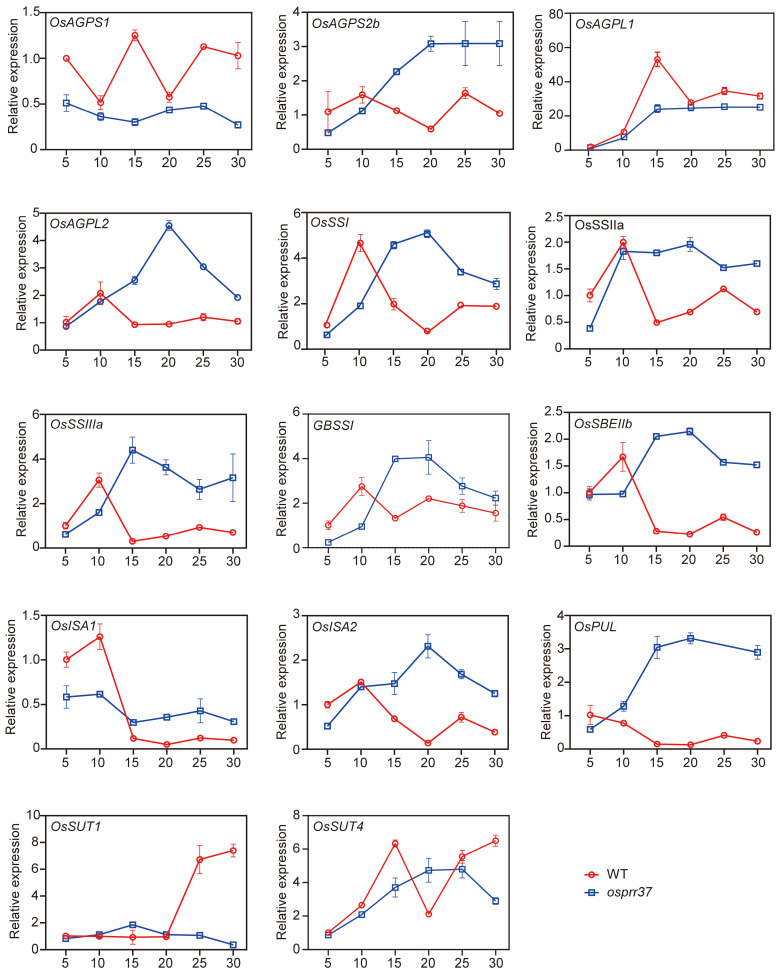
Relative expression of starch synthesis-related genes in wild-type and *osprr37* mutants at various grain filling stages. *OsAGPS1*, *OsAGPS2b* are small subunit genes of ADP-glucose pyrophosphorylase; *OsAGPL1*, *OsAGPL2* are large subunit genes of ADP-glucose pyrophosphorylase; *OsSSI*, *OsSSIIa*, *OsSSIIIa* are starch synthase genes; *OsGBSSI*, granule-bound starch synthase gene; *OsSBEIIb*, starch branching enzyme IIb gene; *OsISA1*, *OsISA2* are starch debranching enzyme genes; *OsPUL*, pullulanase-type of a debranching enzyme gene; *OsSUT1*, *OsSUT4* are sucrose transporter genes. RNA was isolated from developing grains at 5, 10, 15, 20, 25, and 30 DAF. Data indicate means ± SD from 3 biological replicates.

**Figure 10 plants-14-03690-f010:**
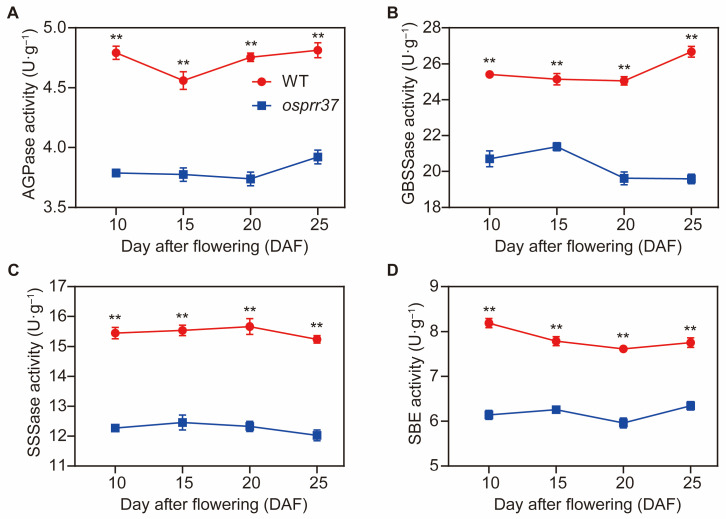
The differences of enzyme activities related to starch synthesis. (**A**) Activity of AGPase (ADP-glucose pyrophosphorylase), (**B**) GBSSase (granule-bound starch synthase), (**C**) SSSase (soluble starch synthase), and (**D**) SBE (starch branching enzyme) in the wild-type and *osprr37* mutants at 10, 15, 20, and 25 DAF. Data indicate means ± SD from 3 biological replicates. ** *p* < 0.01.

**Table 1 plants-14-03690-t001:** Grain filling parameters of grains.

Genotype	Tmax·G (Day)	Gmax (g·1000 Grains^−1^·Day)	D (Day)	$\bar{G}$ (g·1000 Grains^−1^·Day)	A (g·1000 Grain^−1^)
WT	6.507 ± 0.204	2.483 ± 0.111 *	13.303 ± 0.426	1.498 ± 0.058 *	22.834 ± 0.909 *
*osprr37*	6.456 ± 0.238	2.069 ± 0.119	14.324 ± 1.43	1.244 ± 0.072	19.742 ± 0.933

Note: tmax·G, the data of reaching the maximum grain filling rate; Gmax, the maximum grain filling rate; D, active grain filling duration; $\bar{G}$, average grain filling rate; A, final grain weight. Data were analyzed using *t*-test and presented as means ± SD from 3 biological replicates. * *p* < 0.05.

**Table 2 plants-14-03690-t002:** Effect of *OsPRR37* on the diameter of starch granules.

Genotype	Mean Diameter (μm)	Distribution of Volume-Weighted Mean Diameter (%)			Surface Area-Weighted Mean Diameter D [3,2] (μm)	Volume-Weighted Mean Diameter D [4,3] (μm)
		≤2 μm	2–5 μm	≥5 μm		
WT	4.87 ± 0.03 *	6.13 ± 0.41	38.02 ± 0.30	55.85 ± 0.71 *	4.12 ± 0.05	5.05 ± 0.02 *
*osprr37*	4.73 ± 0.01	6.40 ± 0.47	41.02 ± 0.30 **	52.58 ± 0.17	4.02 ± 0.04	4.95 ± 0.02

Data indicate means ± SD from 2 biological replicates. Student’s *t*-test was used to generate the *p* values; * and ** indicate *p* < 0.05 and *p* < 0.01, respectively.

**Table 3 plants-14-03690-t003:** Starch DSC characteristic values of wild-type and *osprr37* mutants.

Genotype	*T_o_* (°C)	*T_p_* (°C)	*T_c_* (°C)	Δ*H* (J·g^−1^)
WT	58.46 ± 0.36	67.62 ± 0.26 *	75.66 ± 0.28	12.42 ± 0.21 *
*osprr37*	57.65 ± 0.37	65.18 ± 1.37	73.82 ± 1.59	10.82 ± 0.82

Data indicate means ± SD from 3 biological replicates. Student’s *t*-test was used to generate the *p* values; * *p* < 0.05.

**Table 4 plants-14-03690-t004:** Starch RVA profile characteristic values of wild-type and *osprr37* mutants.

Genotype	Peak Viscosity PV (cP)	Trough Viscosity TV (cP)	Breakdown BD (cP)	Final Viscosity FV (cP)	Setback SB (cP)	Pasting Temperature PT (°C)	Peak Time PeT (min)
WT	1587.5 ± 24.7 **	947 ± 15.6 **	640.5 ± 9.2 **	1750.5 ± 27.6 **	803.5 ± 12 **	76.6 ± 0.1	6.17 ± 0.05 *
*osprr37*	1142 ± 7.1	701 ± 12.7	441 ± 19.8	1379.5 ± 17.7	678.5 ± 4.9	92.9 ± 0.1 **	6.03 ± 0.05

Data indicate means ± SD from 2 biological replicates. Student’s *t*-test was used to generate the *p* values; * and ** indicate *p* < 0.05 and *p* < 0.01, respectively.

## Data Availability

The original contributions presented in this study are included in the article/Appendix A. Further inquiries can be directed to the corresponding author.

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
