# Peer review of "The Effects of Photoperiodic Transcription Factor OsPRR37 on Grain Filling and Starch Synthesis During Rice Caryopsis Development"

_plants, 2025, doi:10.3390/plants14233690_

Round 1

Reviewer 1 Report

Comments and Suggestions for Authors

Reviewer Comments

The manuscript entitled “The effects of photoperiodic transcription factor OsPRR37 on grain filling and starch synthesis during rice caryopsis development” investigates the regulatory role of OsPRR37 in starch metabolism and rice grain quality. The topic is timely and of potential interest to both plant physiology and crop quality research communities. The authors provide valuable phenotypic, biochemical, and transcriptional data. However, before this work can be considered for publication, I recommend major revision to address several issues related to data presentation, methodological clarity, and mechanistic interpretation.

1. Figure 9 presents the relative expression profiles of starch synthesis-related genes (SSRGs) in WT and osprr37 mutants during grain filling (5–30 DAF). The current heatmap format is suitable for screening differentially expressed genes but insufficient for depicting temporal expression trends of key genes (e.g., OsAGPS1/OsAGPL1 consistently reduced in osprr37, and OsSSI/OsGBSSI/OsSBEIIb upregulated in mid-to-late stages). For example, it is not visually intuitive whether OsAGPS1 expression is stably lower than WT at every stage, or whether OsSSI upregulation begins specifically after 15 DAF. Supplementary line graphs are strongly recommended to clarify temporal expression dynamics and reinforce the transcriptional regulation claims.

2. The study relies exclusively on qRT-PCR, which measures transcript levels but not enzyme functionality. Enzyme activities of AGPase, SS, GBSSI, and SBEIIb are central to validating conclusions (e.g., that reduced OsAGPS1/OsAGPL1 expression impairs ADP-glucose synthesis). Since translation efficiency, post-translational modifications, and metabolite feedback can decouple mRNA levels from activity, enzyme assays are necessary to confirm functional compensation and identify rate-limiting steps.  Measuring the enzyme activities is essential to substantiate the link between transcriptional changes and starch phenotypes.

Author Response

Comment 1: Figure 9 presents the relative expression profiles of starch synthesis-related genes (SSRGs) in WT and osprr37 mutants during grain filling (5–30 DAF). The current heatmap format is suitable for screening differentially expressed genes but insufficient for depicting temporal expression trends of key genes (e.g., OsAGPS1/OsAGPL1 consistently reduced in osprr37, and OsSSI/OsGBSSI/OsSBEIIb upregulated in mid-to-late stages). For example, it is not visually intuitive whether OsAGPS1 expression is stably lower than WT at every stage, or whether OsSSI upregulation begins specifically after 15 DAF. Supplementary line graphs are strongly recommended to clarify temporal expression dynamics and reinforce the transcriptional regulation claims.

Response: We thank the reviewer for the suggestion. We agree that line graphs can more effectively depict the temporal dynamic expression of starch synthesis- related genes. Accordingly, we have replaced the original heatmap with new line graphs as the revised Figure 9. The new line graphs clearly shows that the expression of several starch synthesis genes (e.g. OsAGPS2b, OsAGPL2, OsSSI, OsSSIIa, OsSSIIIa, OsGBSSI, OsSBEIIb, OsISA1, and OsISA2) upregulated in osprr37 during mid-to-late developmental stages. This change has been made on page 13 of revised manuscript.

Comment 2: The study relies exclusively on qRT-PCR, which measures transcript levels but not enzyme functionality. Enzyme activities of AGPase, SS, GBSSI, and SBEIIb are central to validating conclusions (e.g., that reduced OsAGPS1/OsAGPL1 expression impairs ADP-glucose synthesis). Since translation efficiency, post-translational modifications, and metabolite feedback can decouple mRNA levels from activity, enzyme assays are necessary to confirm functional compensation and identify rate-limiting steps. Measuring the enzyme activities is essential to substantiate the link between transcriptional changes and starch phenotypes.

Response: We are grateful to the reviewer for this suggestion. We have now performed enzyme activities assays for ADP-glucose pyrophosphorylase (AGPase), granule-bound starch synthase I (GBSSI), soluble starch synthase (SSSase), and starch branching enzyme (SBE) in developing grains of both wild-type and osprr37 mutants.

The results confirm our initial conclusion that starch biosynthesis was impaired in osprr37 mutants. And we observed significantly reduced activities of four enzymes in osprr37 mutants. The activity of AGPase, the rate-limiting enzyme, is lower in osprr37 mutants, consistent with the downregulation of its encoding genes (OsAGPS1 and OsAGPL1). This directly limit the substrate supply for downstream reactions, explaining the low activities of GBSSI, SSSase, and SBE during early grain filling stages.

The observed upregulation of downstream genes (OsAGPS2b, OsAGPL2, OsSSI, OsSSIIa, OsSSIIIa, OsGBSSI, OsSBEIIb, OsISA1, and OsISA2) in the mid-to-late stages is a key finding, which we interpret as a delayed, compensatory response likely triggered by accumulated sugar signals. However, this compensatory attempt is ultimately ineffective, because the mid-to-late stage is not the optimal period for rapid starch accumulation. Thus the initial downregulation was decisive, it delayed the initiation of starch synthesis and altered starch composition, leading to the reduced yield and quality.

To incorporate these findings, we have made major additions to the manuscript. A new Figure 10 has been added, presented the enzyme activities profile. A subsection in the Results section titled “2.9. The expression pattern of starch synthesis-related genes and enzyme activities” (line 347-356 ) has been included to describe these findings. The discussion section has been updated to integrate these new date with our phenotypic and transcriptional results (page 15).

Reviewer 2 Report

Comments and Suggestions for Authors

Grain filling governs grain weight formation in rice. In this study, the authors focus on the role of OsPRR37 in regulating grain development, starch metabolism, and starch physicochemical properties, systematic comparative analysis of the agronomic traits, grain filling process, expression of starch synthesis-related genes (SSRGs), and starch physicochemical properties of osprr37 mutants and WT, which is sufficient and reliable experimental evidence to reveal the regulatory function of OsPRR37 in rice grain development. The study design is reasonable and the logic is clear, and is a very interesting and practical research which should be seen by the reader of Plants. 

Author Response

Comment: Grain filling governs grain weight formation in rice. In this study, the authors focus on the role of OsPRR37 in regulating grain development, starch metabolism, and starch physicochemical properties, systematic comparative analysis of the agronomic traits, grain filling process, expression of starch synthesis-related genes (SSRGs), and starch physicochemical properties of osprr37 mutants and WT, which is sufficient and reliable experimental evidence to reveal the regulatory function of OsPRR37 in rice grain development. The study design is reasonable and the logic is clear, and is a very interesting and practical research which should be seen by the reader of Plants.  

Response: We are grateful to the reviewer for this positive and encouraging assessment of our work.  Thanks for your supportive comments.

Reviewer 3 Report

Comments and Suggestions for Authors

Dear author,

This manuscript is well-structured. Please follow the ms file

Author Response

Comment 1: In materials and methods section, Please mention specific objectives of each methodology.

Response: We thank the reviewer for this suggestion. We agree that stating the specific for each methodology provide the scientific rationale of experimental design.

Accordingly, we have revised the Materials and Methods section to supplement a statement of purpose for each experimental procedure. Below are several examples that illustrate the modification we have made:

  1. Determination of agronomic traits and grain filling rate: “To elucidate the the process of rice yield formation, we systematically measured key agronomic traits at maturity, ...”
  2. Determination of carbohydrate content: “To elucidate the the physicochemical basis of yield and quality formation, we quantified sugar and starch conent according to the method of Wang et a”
  3. Analysis of starch physicochemical properties: “To analyze key stach composition and function properties, we measured the amylose content, gel consistency, starch water solubility, and swelling power.”

Comment 2: I recommend to draw one graphical abstract reflecting the result and the discussion part.

Response: We thank the reviewer’s suggestion and agree that graphical abstract will enhance accessibility of our study. We have drawn a graphical abstract that summarizes the results and main conclusion of our study.

This graphical abstract summarizes the relationships between storage substrates and rice quality traits: (1) Nutritional quality: protein content is negatively correlated with starch content, especially albumin and prolamin; (2) Appearance quality: chalky grain rate is positively correlated with albumin content, but negatively correlated with starch content, gel consistency, and starch order. (3) Processing quality: whole head rice rate correlated positively with starch content and gel consistency, but negatively with pasting temperature. (4) Eating and cooking quality: determined by a complex interplay where amylose content and starch structure (e.g. order, relative crystallinity) influence pasting, thermal properties. These results demonstrate that the the accumulation of storage assimilates is a critical determinant of overall rice quality.

Comment 3: In conclusion part research gap and future prospect is missing, please mention it

Response: We thank the reviewer for this suggestion. We have revised conclusion section to explicitly state the research gap and future research prospects in our study. The newly added text is as follow:

“Despite we reveal the role of OsPRR37 in regulating grain yield and quality, the precise molecular mechanisms underlying these effects remain largely elusive. Future research should should focus on the following aspects: protein-protein interaction screening using yeast two-hybrid or Co-immunoprecipitation assay could identify functional proteins that binding to OsPRR37 and modulate its expression or activity; high-throughput technologies such as Chip-seq or DAP-seq could be employed to identify its downstream target genes. These approaches will elucidate the molecular mechanisms through which OsPRR37 regulates grain yield and quality.”

This paragraph has been incorporated into the conclusion section on page 21 (line 707-715) of the revised manuscript

Reviewer 4 Report

Comments and Suggestions for Authors
  1. Final part of Introduction, lines 87-92. Please state the objectives of the study directly, rather than describing them as actions to be carried out. In their present form, they resemble the Materials and Methods section.
  2. Results, Figure 5D. In the case of images of isolated starch, from my perspective it is not very clear the difference claimed by the authors concerning that "Wild-type granules were predominantly polygonal structure with angular edges and uniform size, whereas osprr37 starch contained a notable proportion of spherical granules with smooth surfaces and non-uniform size distribution” (lines 198-201). " (lines 198-201). Please reconsider the inclusion of Figure 5D in the final version of the document. I believe that its possible exclusion would not affect the quality of the document in any way.
  3. Results, Figures 7H, 7I, 7J, and 7K (nutritional section), please draw the full standard deviation bar, as only the top section of the bar is visible.
  4. Results, lines 249-254. Overall, the mutant resulted in increased grain protein content. Does this mean an increase in overall grain quality? Does this have a similarity with Opaque-2 corn (or QPM corn)? It is suggested that the authors include a statement on this aspect.
  5. Materials and Methods, lines 512-513. Please report accurately the Richards equation. Note that -kt is a superscript of Be, and -1 is a superscript of everything within the round bracket. Same for min-1 in line 571.

Author Response

Comment 1: Final part of Introduction, lines 87-92. Please state the objectives of the study directly, rather than describing them as actions to be carried out. In their present form, they resemble the Materials and Methods section.

Response: We sincerely thank the reviewer for this insightful comment. We agree that the previous formulation was overly descriptive of the methodology. Following the review’s suggestion, we have rewritten the final paragraph of introduction (now lines 91-98 on page 2 and 3) to directly state the objective and scientific aim of our study.

Comment 2: Results, Figure 5D. In the case of images of isolated starch, from my perspective it is not very clear the difference claimed by the authors concerning that "Wild-type granules were predominantly polygonal structure with angular edges and uniform size, whereas osprr37 starch contained a notable proportion of spherical granules with smooth surfaces and non-uniform size distribution” (lines 198-201). " (lines 198-201). Please reconsider the inclusion of Figure 5D in the final version of the document. I believe that its possible exclusion would not affect the quality of the document in any way.

Response: We thank the reviewer for this critical observation and for the suggestion regarding Figure 5D. We agree with the reviewer that the morphological differences in isolated starch granules, as presented in original Figure 5D, may not be sufficiently clear or conclusive to all readers. This is likely due to the method of our starch extraction. The osprr37 mutant grains exhibited a chalky endosperm, primarily of the “white core” type, which resulted from aberrant starch development. However, other regions of the mature grain (e.g. dorsal and ventral areas) remained translucent and contain normally shaped starch granules. Our extraction protocol isolated starch from the entire grain, resulting in a mixture of both normal, polygonal granules from the translucent regions and the abnormal, spherical granules from the chalky areas. Consequently, the image presents a blend of both morphologies, which may influence the perceived contrast. In response, we have removed the original Figure 5D from the manuscript.

Comment 3: Results, Figures 7H, 7I, 7J, and 7K (nutritional section), please draw the full standard deviation bar, as only the top section of the bar is visible.

Response: We thank the review for pointing out this oversight in Figures 7H, 7I, 7J, and 7K. We have corrected Figures 7H, 7I, 7J, and 7K to ensure that the full standard deviation bar are completely visible. The revised figures have been updated in the manuscript on page 10.

Comment 4: Results, lines 249-254. Overall, the mutant resulted in increased grain protein content. Does this mean an increase in overall grain quality? Does this have a similarity with Opaque-2 corn (or QPM corn)? It is suggested that the authors include a statement on this aspect.

Response: We thank the reviewer’s comment and like to clarify this point. Our conclusion that osprr37 mutants exhibit an overall deterioration in grain quality is based on a comprehensive assessment.

While the increased protein content in osprr37 mutants suggests the improvement in nutritional quality, a comprehensive evaluation system of grain quality must also consider processing, appearance, and eating/cooking quality. The reduction in total starch content resulted in powdery and friable grains, which is commercially undesirable and detrimental to both processing and appearance quality. For eating and cooking quality, the decrease in amylose content led to undesirable alterations in pasting and gelatinization properties, which have negative effect on rice stickiness, elasticity, and overall palatability. Therefore, the net effect of the OsPRR37 mutation is a significant reduction in overall grain quality.The maize Opaque-2 (O2) protein is a key regulator of endosperm development and carbon-nitrogen metabolism, whose loss of function leads to significantly reduced starch content and yield. In our study, osprr37 mutants exhibit a parallel phenotype: disrupted starch metabolism, elevated soluble sugars, reduced starch content, increased protein content, and lower yield. These key traits are highly similar to the phenotypes of the maize o2 mutants, suggesting that OsPRR37 may be similar to the O2 protein and play a regulatory role in grain yield and quality formation.And we have added these paragraphs to the Discussion section to address the points.

Comment 5: Materials and Methods, lines 512-513. Please report accurately the Richards equation. Note that -kt is a superscript of Be, and -1 is a superscript of everything within the round bracket. Same for min-1 in line 571.

Response: We thank the reviewer for pointing out these critical typographical errors in our manuscript. We apologize for this oversight. We have corrected both the Richards equation and the unit notation as per the reviewer’s instructions.

  1. For the Richards equation, we have reformatted it. The term “-kt”now appears as the superscript of “Be”, and the exponent “-1” applies to the superscript of everything within the round bracket. The corrected equation, as it now appears in line 573 of the revised manuscript, is: W=A(1+Be-kt)-1/N.
  2. We have corrected “min-1”to “min-1” in line 637 of the revised manuscript.

Round 2

Reviewer 1 Report

Comments and Suggestions for Authors

I recommend accepting this manuscript for publication